# Intra-Storm Pattern Recognition through Fuzzy Clustering

Konstantinos Vantas [ID] and Epaminondas Sidiropoulos *[ID]

Department of Rural and Surveying Engineering, Aristotle University of Thessaloniki,
54124 Thessaloniki, Greece; kon.vantas@gmail.com
* Correspondence: nontas@topo.auth.gr

**Abstract:** The identification and recognition of temporal rainfall patterns is important and useful not only for climatological studies, but mainly for supporting rainfall–runoff modeling and water resources management. Clustering techniques applied to rainfall data provide meaningful ways for producing concise and inclusive pattern classifications. In this paper, a timeseries of rainfall data coming from the Greek National Bank of Hydrological and Meteorological Information are delineated to independent rainstorms and subjected to cluster analysis, in order to identify and extract representative patterns. The computational process is a custom-developed, domain-specific algorithm that produces temporal rainfall patterns using common characteristics from the data via fuzzy clustering in which (a) every storm may belong to more than one cluster, allowing for some equivocation in the data, (b) the number of the clusters is not assumed known a priori but is determined solely from the data and, finally, (c) intra-storm and seasonal temporal distribution patterns are produced. Traditional classification methods include prior empirical knowledge, while the proposed method is fully unsupervised, not presupposing any external elements and giving results superior to the former.

**Keywords:** rainfall; rainstorm events; inter-event time; intra-storm patterns; fuzzy clustering; clustering analysis; clustering tendency; Greece





## 1. Introduction

Knowledge of the temporal and spatial distribution of rainfall is essential both for climatological studies, especially regarding climate change, and for purposes of flood studies and water resources planning. Effective and illuminating studies of rainfall data are achieved through the detection of patterns or groupings. Stochastic precipitation models utilize Markov chains to simulate the occurrence of wet or dry days and consequently the daily precipitation depth [1–3]. Numerous models, extensively developed, are based on the concept of rectangular pulses point process (RPPP), which can be categorized into two types, the Bartlett–Lewis [4] and the Neyman–Scott [5,6] precipitation models. Both of them use the assumption that storms arrive according to a Poisson process, the most basic example of continued-time Markov chains, of which a concise description can be found in Onof et al. [7]. The application of RPPP methods concerns the fitting of a small set of parameters that define the distributions that follow different rainfall quantities, such as the rainstorm and rain-cell origins, durations and intensities, a multi-objective problem without analytical solution and, sometimes, erratic results [5,7–9].

A different family of models are the profile-based ones [10,11] that utilize the internal structure of rainstorms in their entirety and not as derived parameters of statistical distributions. These rainfall models employ the construction of Huff or mass curves [12], a probabilistic method for sub-daily precipitation records, in which rainstorm data are represented in the form of normalized time, versus normalized cumulative precipitation depth and classified by the quartile where the maximum intensity occurs. The use of mass curves offers several advantages: (a) adequate stability regarding the sample size of rainstorms [13]; (b) coarser hourly data giving nearly identical results to data with

finer time-step in the order of minutes [13]; (c) similarity in very long distances [12–14]; (d) the fact that they are not affected by elevation, type of storm, storm duration or storm precipitation depth [15].

Given that precipitation records contain both wet and dry periods, an important issue is the delineation or the extraction of individual rainstorm events from the contiguous recorded precipitation data. A systematic review [16] of different methods that are based on various rainstorm events characteristics (depth, intensity, generated runoff) reveals the necessity of selecting a criterion that defines objectively the rainless intervals, or minimum inter-event time, and not the arbitrary selection of constant time length. The latter is a common practice: Huff used 6 h to separate storms [12], Yu et al. used 2 h [17] and the calculation of rainfall erosivity in the universal soil loss equation (USLE) and its revisions [18] uses, also, 6 h. A different approach was proposed by Restrepo-Posada and Eagleson [19], commonly used today by many authors [13,20], in which "an empirical and inexact", but easily applied, method was developed for the separation of precipitation timeseries, into statistically independent rainstorms.

Unsupervised learning methods, clustering in particular, are now employed for pattern recognition in hydrological data [21]. In general, clustering analysis is a popular exploratory task aimed at partitioning the content of databases into smaller groups on the basis of inherent similarity criteria [22]. Clustering yields meaningful patterns to be further utilized for understanding of processes or for simulations. Clustering methods are applied in many fields, motivated by the large amounts of accumulated data and the developed needs for better data management through grouping and detection of patterns. The unsupervised nature of clustering analysis is a definite advantage, but it raises additional difficulties regarding suitable choice of methods and metrics. The challenge of clustering also lies in adapting the method and its parameters to the nature of the specific application.

A variety of such unsupervised learning methods are reviewed by Sheikholeslami et al. [23], including descriptions of clustering approaches. Among these, the well-known k-means and k-medoids methods divide the data into distinct clusters, such that each element belongs to exactly one cluster. This type of clustering may be characterized as hard, crisp or non-fuzzy. On the contrary, in fuzzy clustering, each point-element is allowed to belong to more than one cluster, albeit with a varying degree of membership. The membership degree of an element is a number in the interval [0,1]. Points closer to the center of a cluster are assigned a higher membership degree than points further away. One of the most widely used fuzzy clustering algorithms is fuzzy c-means clustering. It is used frequently in pattern recognition, and a related review can be found in Nayak et al. [24]. A major problem in clustering is the determination of the optimal number of clusters, which, in real-world applications, is generally not known in advance. As a result, a number of methods have been developed for the determination of the optimal number of clusters. Many of them use the concept of relative cluster validity [25], where results from different clustering methods are compared using a predefined metric. A number of these methods can be found in Milligan et al. [26] and Charrad et al. [27].

In hydrology, machine learning and clustering methods are being applied with increasing frequency, but, apparently, not to their full potential yet. More specifically, cluster analysis has been applied for the identification of hydrologically homogeneous regions, as noted in several literature publications: (a) hierarchical clustering on monthly rainfall erosivity density [21]; (b) fuzzy c-means clustering on annual precipitation [28] and annual maximum intensities [29]; self-organizing maps on monthly precipitation [30], while less work has appeared in terms of temporal pattern investigations applying: (a) the AL algorithm on rainstorms mass curves [31]; (b) self-organizing maps on design hyetographs; (c) k-means, also, on rainstorms mass curves [32]. On the other hand, rainfall mass curves have been utilized, as previously reported, in the stochastic generation of rainfall events [10,11,33,34], also as design storms [14,35–37], for the simulation of floods [38,39] and in changes in storm properties due to climate change [40].

In view of the above considerations, this paper aims to investigate the presence of intra-storm temporal patterns using timeseries of rainfall data coming from the Greek National Bank of Hydrological and Meteorological Information. Prior to clustering analysis, a more precise statistical analysis is applied in order to define a seasonal model to aid in the extraction of statistically independent rainstorm events, which has never been presented for Greece. The computational process that follows is a custom-developed, domain-specific algorithm that produces temporal rainfall patterns using common characteristics from the data via fuzzy clustering in which (a) every storm may belong to more than one cluster, allowing for some equivocation in the data, (b) the number of the clusters is not assumed known a priori, but is determined solely from the data and, finally, (c) intra-storm and seasonal temporal distribution patterns are produced. The optimal temporal rainfall distribution curves presented recently by the authors employed hierarchical clustering on principal components and utilized data from a specific water region of Greece [41]. Additionally, a preliminary research that utilizes self-organizing maps and crisp clustering has been presented by the authors in a recent European Geosciences Union General Assembly [42]. The present paper (a) adopts the more general approach of fuzzy clustering, which, to our knowledge, has not been applied so far to rainstorm timeseries; (b) the data utilized extend country wide; (c) the advantages of mass curves mentioned in the literature for different parts of the word are materialized for the case of Greece, regarding regionalization and the effect of elevation. Moreover, the overall result of clustering is verified and compared with the established Huff's classification, which has also never been presented for the country, via visualization effected through non-linear projection and topographic maps that are created using emergent self-organizing maps, a method that can reveal patterns in high-dimensional datasets.

## 2. Materials and Methods

The methodology that was applied in the study is presented in Figure 1 in the form of a flowchart. Precipitation data with a time step of 30 min were imported; rainstorms were extracted from the dataset and preprocessed in order to have an appropriate form as input to clustering analysis. A cluster tendency assessment was applied, so as to examine if clustering is meaningful, and the optimal number of clusters were determined by a custom method that utilizes fuzzy c-means. Additionally, the Huff's curves were compiled for the same dataset. The resulted patterns were visualized using a nonlinear projection of the data in two dimensions, and finally, their characteristics were examined in terms of internal and seasonal structure.

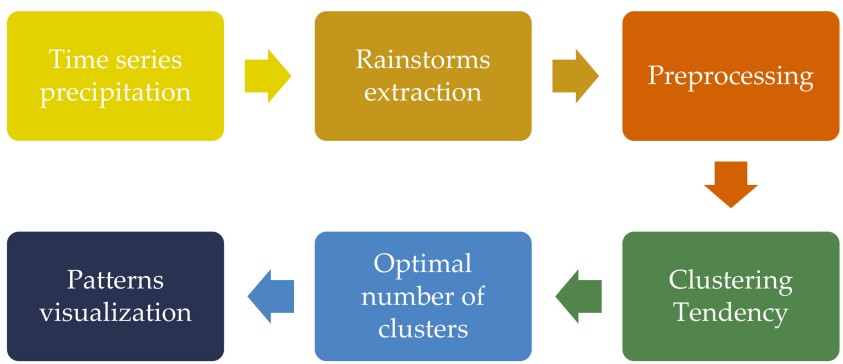

**Figure 1.** Flowchart of the applied methodology.

### 2.1. Data Acquisition and Processing

The data used in the analysis were taken from the Greek National Bank of Hydrological and Meteorological Information [43] for 108 meteorological stations across Greece (Figure 2). The timeseries comprised a total of 2926 years of 30 min records for the time period from 1953 to 1997, with a mean length of 26.6 years per station. The timeseries were

checked for consistency and errors as follows: (a) in records of repetitive values near zero (i.e., ≤0.01 mm), these values were set to zero and (b) records of aggregated values, where the time step was larger than the reported, were removed. The pluviograph data coverage was 43.2% on average. The missing values in the timeseries are not random points but contiguous data related to entire months or years.

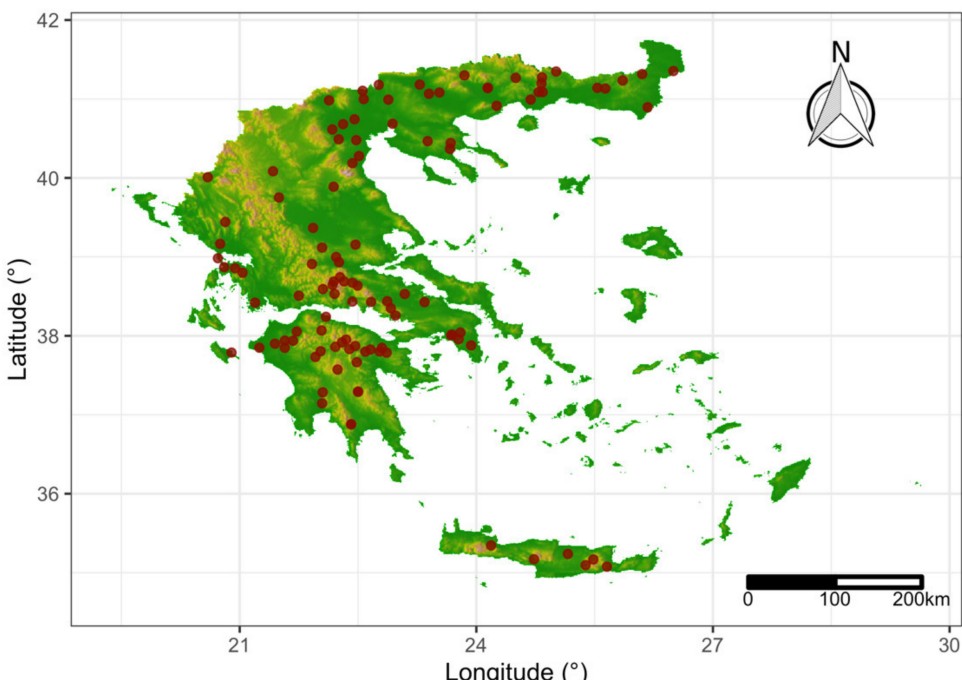

**Figure 2.** Station locations in Greece used in the analysis obtained from the Greek National Bank of Hydrological and Meteorological Information.

Greece, given its geographic characteristics, the distribution of sea and land and the complex and rich relief, has a mosaic of different micro-climates and regional variations, all in the Mediterranean context. Precipitation varies from its maximum values during winter to a minimum during summer. The highest values are observed westwards and on the mountain range of Pindos and its expansion on Peloponnesus (two to three times higher). This fact reveals the importance of relief on the distribution of rainfall over the country. Furthermore, higher precipitation depth is connected to the movement of Mediterranean depressions that follow a characteristic path from west to east. Northerly, at the valleys of central Macedonia, smaller amounts of precipitation are recorded, due to prevailing dry winds. At the northeastern part, at Thrace, higher values are observed also due to relief. Finally, eastern parts of Greece are drier. During summer months when convective activity prevails and over northern Greece, it produces higher precipitation amounts than in the drier southern parts.

### 2.2. Extraction of Rainstorms

A Poisson process hypothesis is assumed for the separation of the precipitation timeseries to statistically independent rainstorm events, where:

- The time intervals of rainstorms that come from the same month are distributed exponentially.
- The rainstorms are separated by a minimum critical time duration of no precipitation (**CD**) or inter-event time [19].
- There is a seasonal pattern for **CD** that is assumed to have constant monthly values.

- The probability density function is:

$$f(t_a) = \omega \cdot e^{-\omega \cdot t_a}, \quad t_a \geq 0 \tag{1}$$

where $\omega$ is the average storm arrival rate and

$$t_a = t_r + t_b \tag{2}$$

where $t_a$ is the storm interarrival time, $t_r$ is the storm duration, and $t_b$ is the dry time between storms.

The estimation of **CD** that separates rainstorms is based on an iterative procedure of statistical tests (Algorithm 1). Inter-month data per station are used to ensure homogeneity and: (a) a test value of *cd* is used, coming from a predefined vector of values **CD**, (b) a vector **T** $_a$ of $t_a$ values is computed for each *cd*, (c) $\omega$ values for **T** $_a$ are estimated by means of the maximum likelihood estimation method, (d) the goodness-of-fit between **T** $_a$ and the exponential distribution is estimated via parametric bootstrapping of s samples that utilizes the Kolmogorov–Smirnov test [44], and (e) the *cd* value with the maximum *p*-value from the empirical non-parametric distribution is selected. In Algorithm 1, a threshold of 50 $t_a$ values was imposed empirically prior to statistical testing.

---

**Algorithm 1:** Temporal model of CD

**Input:** Stations' precipitation time series $P_i$ where $i = 1, \ldots, k$; Critical durations test vector
$\quad$ **CD** $= [120, 180, \ldots, 1800]$ (min); Number of samples that are drawn for parametric
$\quad$ bootstrapping $s = 50,000$;

1   **for** *station* $i \leftarrow 1$ *to* $k$ **do**
2     **for** *month* $m \leftarrow 1$ *to* 12 **do**
3       **for** *cd in* **CD** **do**
4         Compute the vector of interarrival times $t_\alpha$ using inter-month data and $n = \text{length}(t_\alpha)$;
5         **if** $n \geq 50$ **then**
6           Estimate the average storm arrival rate $\hat{\omega}$ from $t_\alpha$ using Maximum Likelihood
            Estimation;
7           Obtain the Kolmogorov–Smirnov's p-value for the original sample $t_a$ and the
            estimated distribution;
8           Generate $s$ samples of size $n$ from the estimated distribution;
9           For each sample compute the one-sample Kolmogorov–Smirnov's p-value using the
            estimated distribution as theoretical;
10          Use the empirical non-parametric distribution of p-values to obtain the p-value for
            the original sample $t_\alpha$;
11     Get minimum dry period duration $MDPD_{i,m}$ from **CD**$[max(p - value)]$;

**Result:** Monthly values of *CD*

---

A first version of the above process was presented in previous publications of the authors [41,45]. The basic hypothesis concerns the application of the probability density function (1). A recent related publication [20] adopted the same exponential distribution to estimate inter-event times in a region of China, but using the "inexact" Restrepo-Posada and Eagleson approach.

## 2.3. Preprocessing

Prior to clustering analysis, preprocessing of data is necessary, in order to transform them into standardized uniform representations that will facilitate the recognition of the common features. More specifically, each one of the storms is a vector of different length, so that a method must be applied to transform the dataset to one with a fixed number of features that represent time. Thus, the storms were scaled and interpolated to unitless

form in which (a) the time expresses the ratio of the rainstorm duration and (b) the rainfall expresses, also, the ratio of total rainstorm depth:

$$p'_i = \frac{p_i}{\sum_{i=1}^n p_i} \tag{3}$$

$$t'_i = \frac{t_i}{\sum_{i=1}^n t_i} \tag{4}$$

where $p_i$ and $t_i$ are the precipitation depth and time for the timestep $i$, respectively, and $p'_i$ and $t'_i$ their scaled values.

In the sequel, since the scaled rainstorm vectors have variable length, linear interpolation was applied to compute the unitless rainfall for every 1% of unitless time values. Finally, a matrix of unitless rainstorms **U** was produced with the values of unitless rainfall, in which every row represents a storm and each column the unitless time values. In the clustering analysis, use was made only of the rainstorms with cumulative rainfall greater than 12.7 mm (i.e., with the exception of light rain) that were no single points, due to the timestep of 30 min, and also erosive by definition [18], as in other similar publications [12–14].

### 2.4. Clustering Tendency

Before the application of a clustering algorithm, it is advisable to have a preliminary look into the dataset in order to detect any existing clustering tendency. This was done in the following ways: (a) a visual assessment of cluster tendency (VAT [46]) and (b) application of the Hopkins index, $H$ [47].

The VAT method creates an image matrix that can be used to visually assess the cluster tendency. In the method, as it was applied, the pairwise dissimilarity values of the rainstorms were computed, and these values were reordered in the form of a matrix and displayed as an intensity image. In this image, the clusters are indicated as dark blocks of pixels along the diagonal [46]. As a dissimilarity measure, the Euclidean distance $d$ was used

$$d(u, v) = \sqrt{\sum_{i=1}^{100} (u_i - v_i)^2} \tag{5}$$

where $u$ and $v$ are two row vectors from the unitless rainstorms **U** matrix. The re-ordering was achieved applying agglomerative hierarchical clustering using the Ward's minimum variance criterion, an algorithm that minimizes the total within-cluster variance [48]. At the beginning of the algorithm, the number of the clusters is equal to the number of data points (all clusters contain a single point). At every step, the algorithm finds the pair of clusters that result after merging to the minimum increase in the total within-cluster variance, which is expressed as the sum of squared differences between the clusters' centers. Finally, all clusters are combined to one cluster that contains all the data using a hierarchical method [21].

The Hopkins index, $H$, can be used to test the null hypothesis of randomly and uniformly distributed data, generated by a Poisson point process and is calculated with

$$H = \frac{\sum_{j=1}^m u_j^d}{\sum_{j=1}^m w_j^d + \sum_{j=1}^m u_j^d} \tag{6}$$

where $X$ is a collection of $n$ data points that have $d$ dimensions. A random sample from $X$ without replacement with members $x_i (i = 1 \text{ to } m, m \ll n)$ is formed. $Y$ is a set of uniformly random data points, also with $d$ dimensions and members $y_j (j = 1 \text{ to } m)$, $u_j$ in turn is the Euclidean distance from $y_j$ to its nearest neighbor in $X$, and $w_j$ is also the Euclidean distance from $x_i$ to its nearest neighbor in $X$. A value of $H$ close to one indicates that the data are highly clustered, 0.5 indicates randomly distributed data, and zero indicates regularly spaced data [21,25].

### 2.5. Fuzzy Clustering

The unitless rainstorm data were clustered by the fuzzy c-means (FCM) algorithm, which was developed by J. Dunn [49] and improved by J. Bezdek [50]. FCM aims to minimize the objective function

$$\underset{C}{\text{argmin}} \sum_{i=1}^{n} \sum_{j=1}^{c} w_{ij}^{m} \|x_i - c_j\|^2 \tag{7}$$

where $w_{ij} \epsilon [0,1]$ is the degree or membership of item $x_i$ from a set of $n$ elements $X = x_1, \ldots, x_n$ that belong to cluster $c_j$, $C$ is the set of $c$ cluster centers $C = c_1, \ldots, c_c$, and $\| \ldots \|$ denotes any norm that expresses similarity, such as the Euclidean distance.

From an iterative optimization procedure on (7), each element $w_{ij}$ of the partition matrix $W$ is equal to

$$w_{ij} = \frac{1}{\sum_{k=1}^{c} \left( \frac{\|x_i - c_j\|}{\|x_i - c_k\|} \right)^{\frac{2}{m-1}}} \tag{8}$$

where $m$ is the "fuzzifier" that determines the level fuzziness with $m \epsilon \mathbb{R}$ and $m \geq 1$, commonly set to two [51]. The larger the $m$ value is, the fuzzier the membership values of the clustered data points are. The centers $c_j$ of the clusters are the mean of all the elements weighted by their degrees:

$$c_j = \frac{\sum_{i=1}^{n} w_{ij}^{m} x_i}{\sum_{i=1}^{n} w_{ij}^{m}} \tag{9}$$

FCM stops when the maximum number of iterations given a priori is reached, or when the algorithm is unable to reduce the current value of the objective function further to a predefined, usually very small, value.

### 2.6. Optimal Number of Clusters

The most common and fundamental problem in clustering analysis is the determination of the number of clusters in an unlabeled dataset, as the one used in the analysis. As previously reported and due to the fact that most clustering algorithms, including FCM, require the number c of clusters to be known a priori, the next step, after answering the question of clustering tendency, has to do with the determination of c. The proposed method uses statistical testing (Algorithm 2). Variations of it have been presented by the authors in the context of different clustering algorithms, namely, (a) self-organizing maps [42] and (b) a dendrogram produced by hierarchical clustering on principal components [41].

A different approach has been presented [52], using k-means (crisp) clustering, which involves two different parameters: (a) an initial threshold that indicates similarity, through a Pearson correlation coefficient, between all pairs of the unitless cumulative rainstorms and (b) an additional one, based on the distances matrix of all pairs of the unitless cumulative rainstorms, which creates the initial seeds used in the k-means clustering algorithm. In other studies, the number of clusters was set without any estimation [32] or through a visual, ambiguous method involving the density maps derived from self-organizing maps [53]. Comparing the reported methodologies to the proposed one, Algorithm 2 presents the following advantages: (a) It is easier to implement, and it can be combined with any clustering algorithm that uses as a parameter the number of clusters directly or indirectly, as is for example the *eps* parameter in the DBSCAN algorithm [54]. (b) It utilizes hypothesis testing about the centers of the clusters, and (c) it has only one parameter to choose, the significance level of well-known statistical tests.

In Algorithm 2, firstly, the cumulative values of the unitless rainstorms matrix are computed, as that kind of transformation was found to help the computational procedures of clustering algorithms. At each step of the iteration, FCM is applied using a trial number of clusters, greater than two and smaller than a predefined maximum value $n_{max}$. Afterwards, the cluster centers that resulted are represented as cumulative rainfall distributions.

These centers, and for all possible pairs, are tested to find out whether they are drawn from the same distribution using the two-sample Kolmogorov–Smirnov test [55]. The test quantifies the distance between the two samples' empirical distribution functions and has been used in similar comparisons [56,57]. Due to the multiple statistical testing that arise, the resulted *p*-values are adjusted using the Benjamini and Hochberg method [58], which controls the false discovery rate. The algorithm stops when any *p*-value is greater than a predefined significance level $\alpha$, and in that case, the clusters from the previous step are returned.

---

**Algorithm 2:** Optimal number of clusters using FCM

---

    **Input:** unitless rainstorms $U$; maximum number of clusters to test $n_{max}$; significance level $\alpha = 0.05$
1  compute the row-wise cumulative values $U'$ of $U$
2  **for** $i \leftarrow 2$ *to* $n_{max}$ *and all p-values* $< \alpha$ **do**
3      apply FCM on $U'$ for $c = i$ and compute cluster centers $C$;
4      for all pairs in $C$ obtain the Kolmogorov–Smirnov two sample test, p-values;
5      adjust the obtained p-values using Benjamini and Hochberg method;
    **Result:** optimal number of clusters $c_{opt}$ and clustering results

---

### 2.7. Projecting Data Using Non-Linear Mapping

The non-linear method that was used to visualize the rainstorm data in a two-dimension scatterplot was the generalized U-Matrix [59]. This method uses the results from a dimensional reduction method, such as principal components analysis, or a non-linear method, such as t stochastic neighbor embedding (t-SNE, [60]), in conjunction with emergent self-organizing maps (ESOM [61]). This step is a necessity due to the Johnson–Lindenstrauss lemma [62] stating that the "low-dimensional similarities do not represent high-dimensional distances coercively" [63]. In other words, specific measures are taken to avoid the common mistake of assuming that, when the projected points are similar to each other after dimensional reduction, such is the case in the actual high-dimensional space. The generalized U-Matrix after its calculation by ESOM is visualized by a three-dimensional space called "topographic map with hypsometric tints" or colors on surface that represent elevation ranges [64]. The topographic map has no actual real borders and is toroidal, which means that the left border is connected to the right one and the top to the bottom.

### 3. Results and Discussion

#### 3.1. Rainstorms Extraction

Empirically, a minimum set of 50 values per month and station were used in Algorithm 1 and some basic statistics; results about CD are presented in Table 1. Due to the dry summer period in Greece, these monthly CD values could be computed only in a small set of stations. A seasonal sinusoidal model (Figure 3) was developed to use in Greece:

$$f(CD) = \theta_1 \sin\left(\frac{2\pi}{12}m\right) + \theta_2 \sin\left(\frac{4\pi}{12}m\right) + \theta_3 \cos\left(\frac{2\pi}{12}m\right) + \theta_4 \cos\left(\frac{4\pi}{12}m\right) \quad (10)$$

where *m* is the month, and $\theta_1, \ldots, \theta_4$ parameters that were estimated using linear regression.

Using the CD sinusoidal model, a set of 174,883 rainstorms were extracted from the dataset. With the exception of summer months and September, the model gives values around 6 h, close to the value selected empirically by Huff [12] and in rainfall erosivity calculations [65]. A subset of 26,678 rainstorms that met the criterion of minimum depth were preprocessed and used to compile the $U$ matrix.

**Table 1.** Average statistical properties of monthly **CD** values for the stations. SD is an abbreviation for standard deviation and CV for coefficient of variation (the ratio of the standard deviation to the mean) and h for hours.

| CD (h) | Min | Mean | Median | Max | SD | Skew | Kurtosis | CV |
|---|---|---|---|---|---|---|---|---|
| January | 2 | 5.4 | 5 | 13 | 1.6 | 1.40 | 4.77 | 0.19 |
| February | 2 | 5.0 | 5 | 10 | 1.4 | 0.90 | 1.39 | 0.16 |
| March | 2 | 5.9 | 5 | 12 | 1.8 | 0.95 | 0.82 | 0.20 |
| April | 4 | 6.3 | 6 | 10 | 1.4 | 0.62 | −0.20 | 0.16 |
| May | 4 | 6.8 | 6 | 12 | 1.9 | 0.91 | 0.49 | 0.24 |
| June | 4 | 8.2 | 8 | 13 | 2.1 | 0.15 | −0.50 | 0.34 |
| July | 5 | 9.3 | 9 | 13 | 2.0 | −0.17 | −0.01 | 0.58 |
| August | 5 | 7.8 | 8 | 11 | 2.1 | 0.12 | −1.73 | 0.70 |
| September | 6 | 9.1 | 9.5 | 11 | 1.6 | −0.52 | −1.03 | 0.45 |
| October | 2 | 7.4 | 7 | 13 | 1.9 | 0.36 | 0.78 | 0.25 |
| November | 2 | 6.7 | 6 | 11 | 1.6 | 0.25 | 0.17 | 0.19 |
| December | 2 | 5.2 | 5 | 11 | 1.4 | 0.85 | 2.00 | 0.16 |

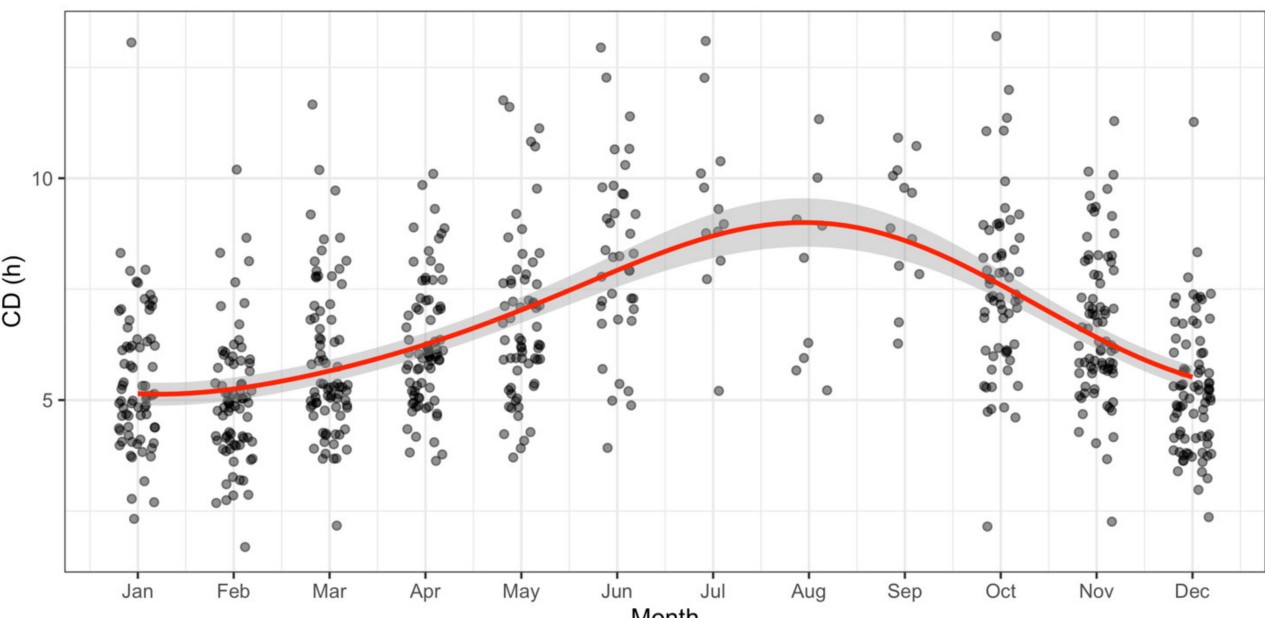

**Figure 3.** Red represents the monthly sinusoidal model of CD (Equation (11)). The grey area corresponds to the standard error of the model using linear regression. The grey dots represent the monthly CD values per station, as computed from Algorithm 1.

### 3.2. Clustering Tedency

Due to computation issues, a random sample of 10% was used to compute the value of the Hopkins index with H = 0.886, so the null hypothesis of random data could safely be rejected. This result indicated that there was a physical meaning in the categorization of the rainstorms based on their internal structure. The VAT method (Figure 4) created an image matrix, where the clusters are indicated as at least four darker blocks along the diagonal. Given the results from these two methods, there is strong evidence that the dataset contains meaningful clusters and the next step can follow, to apply the proposed method and select the optimal number of clusters.

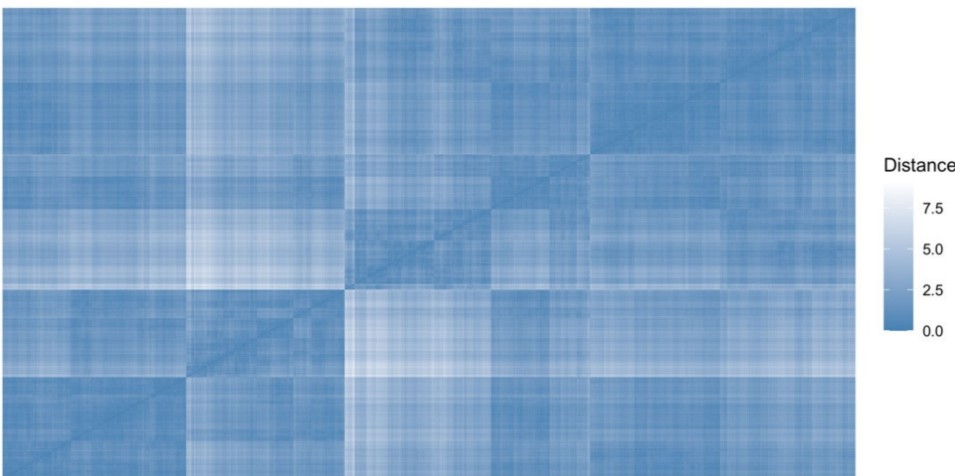

**Figure 4.** Image matrix created by the VAT method, in order to visually inspect the clustering tendency of data. Distances (dissimilarities) among unitless rainstorms are unitless. The rainstorms that belong to the same cluster are displayed in successive order.

### 3.3. Clustering Results and Visualization

After the application of Algorithm 2, the optimal number of clusters are proposed to be four, a suitable number comparing it with VAT results. In Figure 5, the matrix of unitless rainstorms **U** is depicted, with its rows, the rainstorms, reordered by the cluster they belong to. This representation shows that the rainstorms belonging to clusters number one and four are more similar. The latter statement is strengthened by Figure 6 that illustrates the distribution of the membership of every unitless rainstorm that belongs to a cluster (that value must be >0.25 to be classified to a cluster). In the same figure, the mean values of the memberships for each one of the clusters are {0.67, 0.60, 0.59, 0.67}. The latter values are not normally distributed, and cluster one and four are left-skewed meaning less equivocality, in contrast to clusters two and three that are right-skewed.

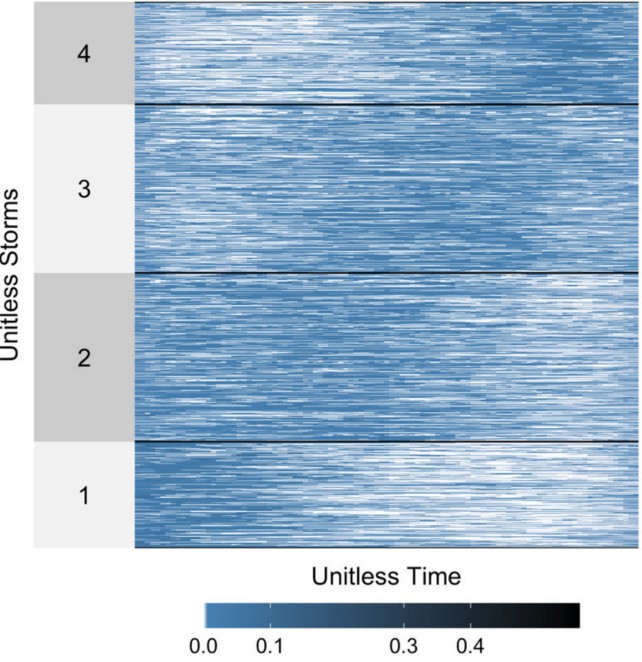

**Figure 5.** Unitless rainstorms values re-order by clustering results. The names of the clusters were used in order to match with Huff's classification. White symbolizes zero precipitation depth, and different blues depict the presence of unitless precipitation.

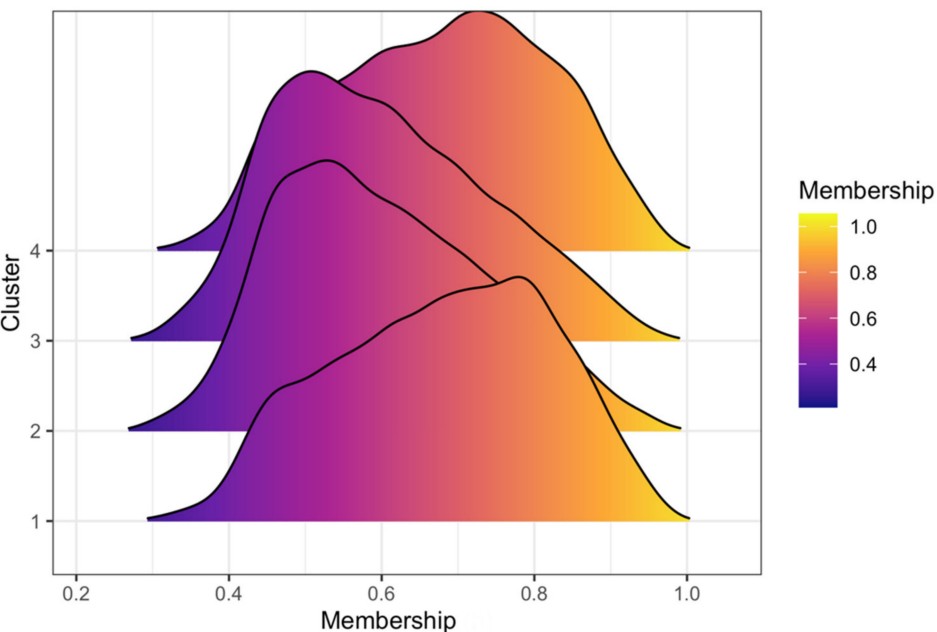

**Figure 6.** Distribution of the membership values from fuzzy c-means (FCM) of every unitless rainstorm that belongs to a cluster.

From Figure 5, initially, it seems that the selection of Huff about the grouping in four clusters has physical meaning, in contrast to criticism about being artificial [66].

The average values related to precipitation depth, duration and intensity per cluster are presented in Table 2. Cluster one has on average shorter duration rainstorms, with lower precipitation depth, but higher mean intensity. Clusters two and three have both rainstorms with similar statistics, with larger duration and precipitation depth than one and four. Cluster four has analogous statistics to one with the exception of intensity, which is lower.

**Table 2.** Average values of occurrence of clusters, duration and precipitation depth intensity of clusters' rainstorms.

|  | Cluster | | | |
|---|---|---|---|---|
|  | **1** | **2** | **3** | **4** |
| Cluster Ratio (%) | 19.5 | 30.86 | 30.84 | 18.79 |
| Duration (h) | 12.5 | 16 | 16.5 | 14.5 |
| Precipitation depth (mm) | 20.7 | 23.2 | 23.7 | 21.9 |
| Intensity (mm/h) | 1.82 | 1.66 | 1.66 | 1.71 |

Cluster number one has notably higher variance and maximum occurrence during spring and summer months that match convective activity over Greece (Figure 7). Clusters two and three have distributions similar to each other and inverse to the first one, related to the prevailing winter weather systems, having their minima during summer. Cluster four has a more uniform distribution with the exception of winter months, when it has its maximum. The latter cluster, with its relatively higher intensity values and, also, higher ratio of occurrence during the wetter winter months in Greece, can be utilized in hydrologic design in the construction of design storms.

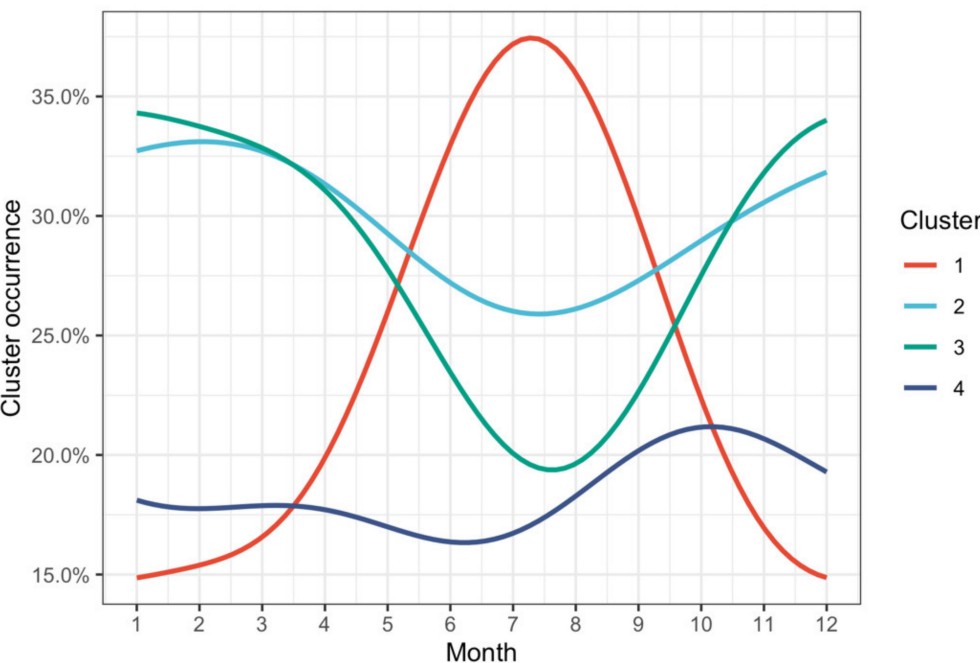

**Figure 7.** Seasonal variability of clusters' monthly occurrence.

In order to examine if elevation affects the distribution of rainstorms in the clusters, density plots were compiled (Figure 8) for each cluster. The distribution of the elevations of the stations related to rainstorms appears to be almost identical, despite the fact that altimetry is closely connected to the precipitation regime in Greece. This result about elevation is in accordance with the reported one for Huff's curves by Loukas et al. in Canada [15].

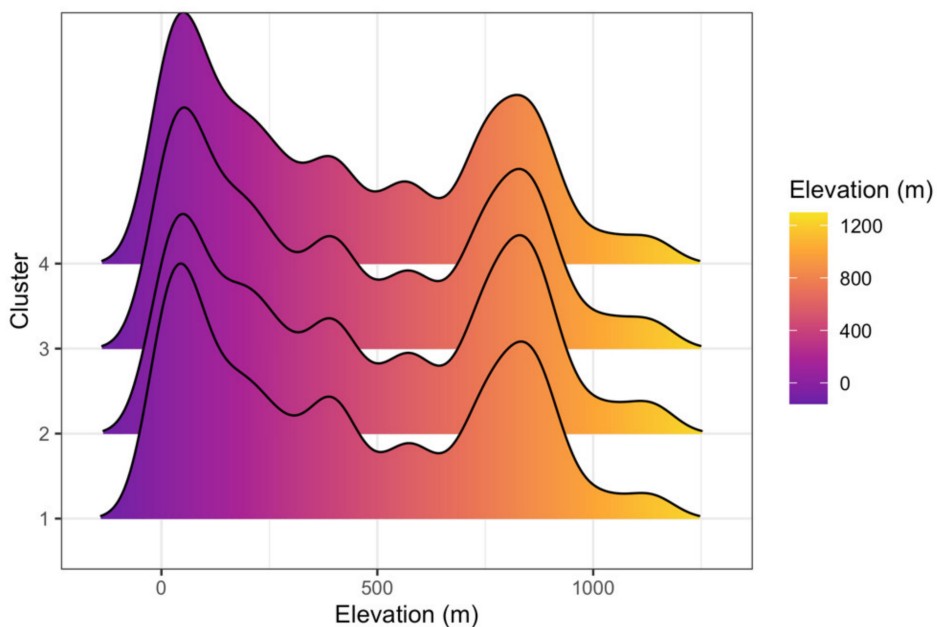

**Figure 8.** Distribution of elevation values from FCM of every unitless rainstorm given its cluster.

In order to examine the temporal patterns in every station using the results from FCM, correlation matrices were computed using Pearson's r coefficient [67]:

$$\text{r} = \frac{1}{n-1} \sum_{i=1}^{n} \left( \frac{x_i - \overline{x}}{s_x} \right) \left( \frac{y_i - \overline{y}}{s_y} \right) \tag{11}$$

where $x_i$ is the average values of unitless rainstorms coming from a station $x$ and cluster $c$, $y_i$, similarly, is the average values of unitless rainstorms coming from a station $y$ and the same cluster $c$, $n = 100$ (the ordinates of unitless rainstorms), $s_x$ and $s_y$ are the standard deviations of $x_i$ and $y_i$ and, finally, $\overline{x}$ and $\overline{y}$ the mean values, respectively. The patterns per station and given cluster showed very high similarity with $\text{r} \geq 0.974$ and on average for all values $\text{r} = 0.999$. These results indicate that despite the rich topography and the variability of micro-climates in Greece, the clusters have regional stability in long distances.

In the sequel (Figure 9), the cumulative values of the centers of the four clusters are compared to the ones calculated using Huff's method of classification that is based on the quartile with the highest intensity. The two methods produce different results, while fuzzy clustering creates unitless cumulative curves that do not overlap. The four Huff curves were also tested, for all possible pairs, whether or not are drawn from the same distribution using the Kolmogorov–Smirnov test and three pairs failed to reject that hypothesis for both significance levels $a = 5\%$ and $a = 10\%$.

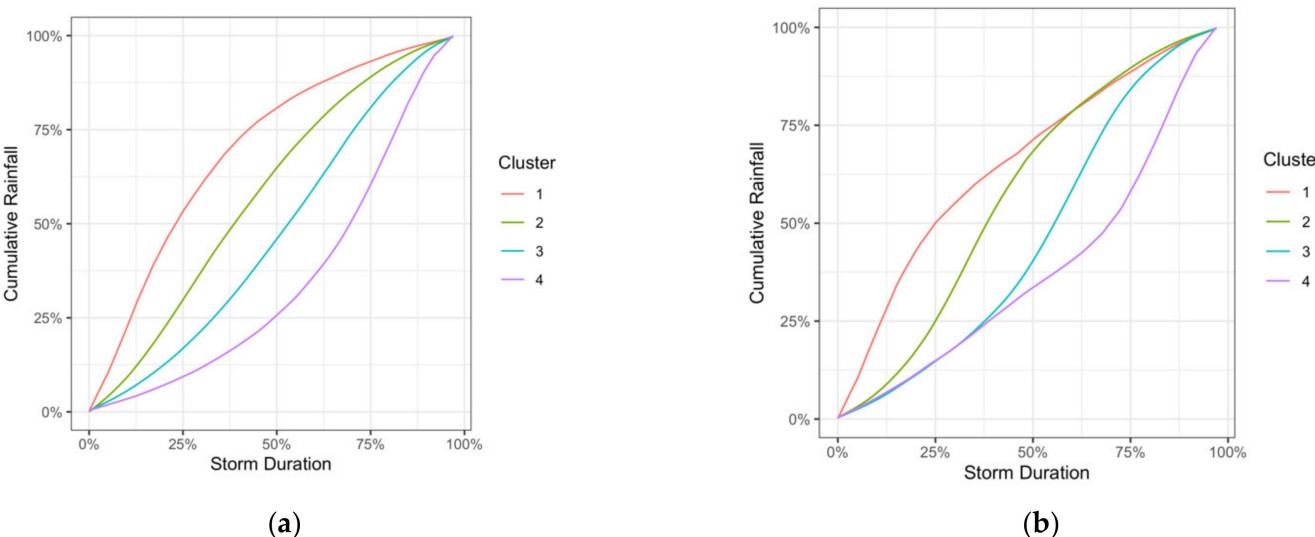

(**a**)　　　　　　　　　　　　　　　　　　　(**b**)

**Figure 9.** (**a**) Unitless cumulative values using FCM. (**b**) Unitless cumulative curves using Huff's classification.

Finally, the topographic maps utilizing the generalized U-matrix using ESOM, as described in Section 2.7, using both FCM and Huff's classification results were produced (Figure 10). In both maps, the unitless rainstorms with membership >80% are used to remove equivocation from the data. The visualization indicates the presence of four clusters and that Huff's method misclassifies the data. The topographic map produced valleys with blue and green color (indicating smaller distances of the points) that are separated by ridges of brown and white color (representing larger distances).

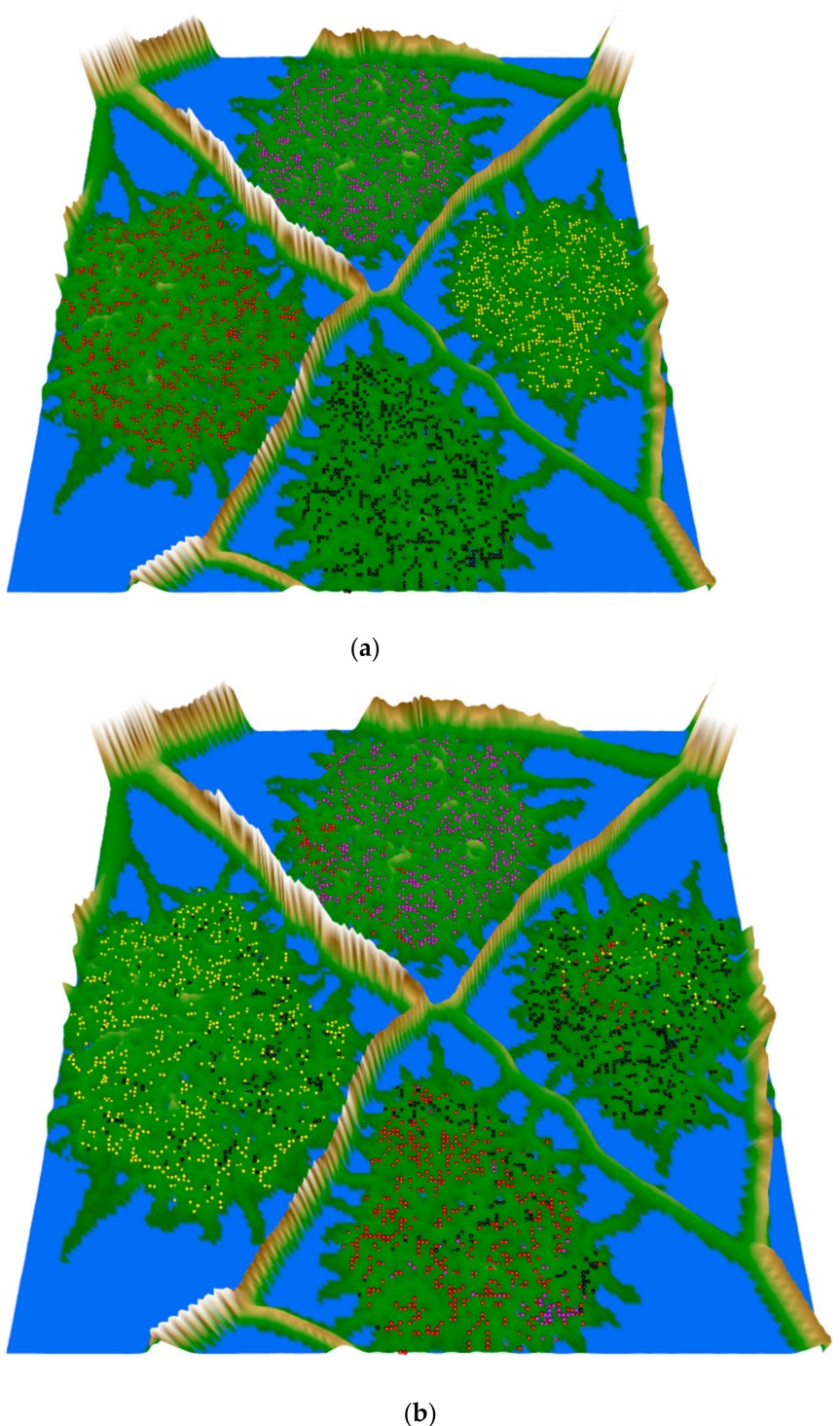

(**a**)

(**b**)

**Figure 10.** Topographic maps with hypsometric tints using t stochastic neighbor embedding (t-SNE) and the generalized U-matrix. The topology of the map is toroidal (the top and bottom and as well the right and left are joined). Each colored sphere represents a rainstorm with its different classification, as it has already computed. (**a**) Results using FCM: the rainstorms that belong to the same cluster are separated by mountain ranges. (**b**) Results using Huff's classification: the rainstorms of different classification are mixed, despite the visual separation of mountain ridges in the map.

## 4. Conclusions

In this paper, the timeseries of rainfall data were processed in order to detect patterns. The data, coming from the Greek National Bank of Hydrological and Meteorological

Information, cover the totality of the country's regions. In the various stages of the project, novel methods were presented and implemented. First, extraction of rainstorms was executed via a stochastic method. Then, the timeseries were properly transformed to unitless form. The data were tested for clustering tendency prior to the application of clustering algorithms. An iterative form of fuzzy clustering was presented that does not assume a priori knowledge of the number of clusters. It consists of a repeated execution of fuzzy c-means clustering in combination with statistical testing for the choice of the relevant number of clusters. Finally, a verification of the clustering results and a comparison to the widely used Huff's method was attained through topographic maps. The overall approach extends previous rainfall pattern recognition efforts of the authors and of the literature in the sense of the unsupervised learning and in the direction of fuzzy analysis in rainfall and in any follow-up study of floods or water management problems.

**Author Contributions:** Conceptualization, methodology and software K.V.; writing—original draft preparation, K.V. and E.S.; writing—review and editing, E.S. All authors have read and agreed to the published version of the manuscript.

**Funding:** This research received no external funding.

**Institutional Review Board Statement:** Not applicable.

**Informed Consent Statement:** Not Applicable.

**Data Availability Statement:** Precipitation data are available from the Greek National Bank of Hydrological and Meteorological Information.

**Acknowledgments:** The data importing, analysis and presentation were done using the open source R language for statistical computing and graphics [68] using the packages: hydroscoper [43], hyetor [69], e1071 [70], FCPS [71], GeneralizedUmatrix [63], factoextra [72] and ggplot2 [73].

**Conflicts of Interest:** The authors declare no conflict of interest.

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
