# Peer review of "Intra-Storm Pattern Recognition through Fuzzy Clustering"

_hydrology, doi:10.3390/hydrology8020057_

Round 1

Reviewer 1 Report

This paper presents a very solid analysis of an unsupervised storm clustering study. There is nothing that I would criticize in what has been presented, but it would be interesting to see a more descriptive interpretation of what the clustering reveals and would suggest some short additions. 

Can the authors please make some comments on what are the characteristics of a storm in cluster 1 that differentiate it from a  storm in cluster 2, and so on? Is that a storm with more rain over a shorter duration, for example? Some of this is shown in Table 2, but not really described in these terms.

Figure 7, is also probably revealing some of this information as it shows a seasonal dependence on the frequency of membership in the clusters. However, this figure is not referred to in the text, and a discussion of it would give further insight into the clustering results.

One other minor point I would make is that I believe that "precipitation height" is more usually referred to as "precipitation depth". 

Reviewer 2 Report

Ms. Claudia Dragan

Assistant Editor

Hydrology MDPI journals

My revised version of the paper "Intra-storm pattern recognition through fuzzy clustering" has been revised according to manuscript number: hydrology-1146812.

This research investigates the time series of rainfall data to identify and extract representative patterns in Greece.  The paper is well written. However, my primary concern is comparing your results with other methods; the authors only applied their method and no established and comparison with other similar methodologies (“fuzzy clustering”). I suggest that the authors present in a table all the (similar) methods available in the literature regarding pattern recognition using fuzzy clustering in the methodology.

The authors concluded that “In the various stages of the project novel methods were presented and implemented”. Nevertheless, as a Hydrology engineer, I suggest to the authors create a numerical example of their methodology in supplementary material concerning the results from other literature methods.

I have detailed some specific comments below: What is the term “(h)” in the Table 1 and Figure 2?

Regards,

Reviewer,

Reviewer 3 Report

The paper has presented an interesting classification method to identify temporal patterns of rainfall storms which are critically important for hydrological applications of rainfall data. The paper is well-written with nice analysis and results. I have only a few minor comments as follows.

  1. I suppose the presented method is an extension of the authors’ previously published works. I suggest specifying the new additions to the method used in this paper.
  2. The use of the Poisson process to separate rainstorm events seems to work very well. However, it would be interesting to discuss its advantages and disadvantages over other commonly used approaches like the Markov Chain process, etc. (see Urdiales et al., 2018, DOI: https://doi.org/10.3390/w10020145; Chowdhury et al., 2019, DOI: https://doi.org/10.1002/joc.6071).
  3. The clusters are not fully normally distributed. Were they transformed to normal?
  4. Figure 7: why is the seasonality of cluster 1 opposite of the seasonality of clusters 3 and 4?
  5. Figure 9 is not clear – please provide appropriate legend and caption.

Reviewer 4 Report

General comments:
The manuscript deals with an interesting topic for the readers and it is well written.
However, I have some concerns about two methodological aspects that are on the basis of the work.

1) In the study, the pluviograph data of stations distributed over a very large territory with different topographical characteristics were processed.
Since the altimetry significantly influences the rainfall regime, I think it is more correct to divide the territory in altimetric bands to be analyzed separately.
Authors should discuss this aspect.

2) The authors should explain more extensively why they have decided to consider only the rainstorms with duration greater than 3 hours (lines 161-163).
Indeed they should clarify if the use of rainfall with shorter duration (and thus high intensity) may modify the application of the proposed method and influence the results.

Specific comments:

lines 10-12: What does "are delineated to independent rainstorms" mean?
line 100: Please, amend "Cluster"
line 202: Amend "where m is the"
line 294: Amend "storms"
lines 311-315: I think these sentences are describing results of Figure 7, but the Figure 7 is not referred in the text.

Round 2

Reviewer 2 Report

Authors have addressed all my comments.

Reviewer 4 Report

Authors answered adequately at all my requests, I do not have any additional comments.